# Single-Cell Protein and Ethanol Production of a Newly Isolated *Kluyveromyces marxianus* Strain through Cheese Whey Valorization

**DOI:** 10.3390/foods13121892

**Published:** 2024-06-16

**Authors:** Danai Ioanna Koukoumaki, Seraphim Papanikolaou, Zacharias Ioannou, Ioannis Mourtzinos, Dimitris Sarris

**Affiliations:** 1Laboratory of Physico-Chemical and Biotechnological Valorization of Food By-Products, Department of Food Science & Nutrition, School of Environment, University of the Aegean, Leoforos Dimokratias 66, 81400 Myrina, Lemnos, Greece; danaikouk@aegean.gr (D.I.K.); zioan@aegean.gr (Z.I.); 2Department of Food Science and Human Nutrition, Agricultural University of Athens, 75 Iera Odos, 11855 Athens, Greece; spapanik@aua.gr; 3Laboratory of Food Chemistry and Biochemistry, Department of Food Science and Technology, School of Agriculture, Aristotle University of Thessaloniki, 54124 Thessaloniki, Greece; mourtzinos@agro.auth.gr

**Keywords:** single-cell protein, cheese whey, bioconversion, bioethanol, amino acids

## Abstract

The present work examined the production of single-cell protein (SCP) by a newly isolated strain of *Kluyveromyces marxianus* EXF-5288 under increased lactose concentration of deproteinized cheese whey (DCW) and different temperatures (in °C: 20.0, 25.0, 30.0 and 35.0). To the best of the authors’ knowledge, this is the first report examining the ability of *Kluyveromyces marxianus* species to produce SCP at T = 20.0 °C. Different culture temperatures led to significant differences in the strain’s growth, while maximum biomass and SCP production (14.24 ± 0.70 and 6.14 ± 0.66 g/L, respectively) were observed in the cultivation of *K. marxianus* strain EXF-5288 in shake-flask cultures at T = 20.0 °C. Increased DCW lactose concentrations (35.0–100.0 g/L) led to increased ethanol production (Eth_max_ = 35.5 ± 0.2 g/L), suggesting that *K. marxianus* strain EXF-5288 is “Crabtree-positive”. Batch-bioreactor trials shifted the strain’s metabolism to alcoholic fermentation, favoring ethanol production. Surprisingly,* K. marxianus* strain EXF-5288 was able to catabolize the produced ethanol under limited carbon presence in the medium. The dominant amino acids in SCP were glutamate (15.5 mg/g), aspartic acid (12.0 mg/g) and valine (9.5 mg/g), representing a balanced nutritional profile

## 1. Introduction

Cheese whey is the main by-product of cheese production and refers to a liquid stream (emanating) stemming from the process of agglomeration of casein micelles (fractions) [1]. Whey has a greenish-yellow color, and regarding acidity, there is segregation due to derivation [2]. Cheese whey is mainly composed of water, containing approximately 50% of milk nutrients, while the dry matter fraction consists of lactose [66–77% (*w*/*w*)], proteins [8–15% (*w*/*w*] and minerals [7–15% (*w*/*w*] [3] and is considered as the most important pollutant of the dairy industry due to its high organic load. Specifically, chemical oxygen demand (COD) of cheese whey varies from 50,000 to 80,000 mg/L, whilst biochemical oxygen demand (BOD) is around 40,000 to 60,000 mg/L [1]. Major volumes of cheese whey are produced annually, considering that 1 kg of cheese production results in 9 L of whey [1]. According to Eurostat [4], Europe produced 55.5 million tons of whey as a by-product of the manufacture of cheese in 2020. Although advanced technologies such as microfiltration and nanofiltration have been applied for the utilization of cheese whey [5,6] with applications in the food industry [7], the remaining deproteinized stream (permeate whey, lactose rich) still consists of high organic load [3]. Therefore, fermentation processes have aroused interest in the utilization of deproteinized cheese whey as a carbon source [8,9,10,11].

Vivid research interest has arisen for *Kluyveromyces marxianus*, a non-conventional yeast species, due to its physiological characteristics and biotechnological applications. *K. marxianus* is one of the known fastest-growth yeast with a maximum growth rate of 0.80 h^−1^ [12], while it is thermotolerant to 52 °C [13] and acid-tolerant to pH 2.3 [14]. Moreover, *K. marxianus* can utilize a plethora of carbon sources, including xylitol and glycerol [15], inulin [16], lactose [17], as well as low-cost substrates, such as cheese whey [18], lemon peels, sugarcane bagasse [19] and rice bran [20]. It is reported that *K. marxianus* strains are Crabtree-negative [21] and aerobic conditions favor respiration over alcoholic fermentation [22]; however, *K. marxianus’s* overall metabolic behavior in glycolysis and the tricarboxylic acid (TCA) cycle, remain unclear requiring further investigation. Due to its extensive use and presence in dairy products, it has gained the Qualified Presumption of Safely (QPS) and “Generally Recognized as Safe” (GRAS) status in the European Union and the United States, respectively. Consequently, many studies have investigated the ability of *K. marxianus* to produce added-value compounds of high biotechnological interest, including bioethanol [23], aroma and flavor compounds [20,24], polyketides [14], polysaccharides such as inulin [25] and single-cell protein (SCP) [17].

Since SCP from fungal species presents a rich nutritional profile containing valuable amino acids with potential applications in feed and human food, it raises interest as an alternative protein source [26]. Deflection of contamination due to high temperatures during culture media fermentation combined with the thermotolerance that characterizes *K. marxianus* has led many studies to conduct experiments under extreme temperatures [14,17,18,27]. However, high fermentation temperature demands intensive energy and therefore, the procedure presents a high cost. To the best of the authors’ knowledge, there is a lacuna in literature on how *K. marxianus* growth could possibly be influenced under lower temperatures. In addition, due to its rapid growth, there is no evidence in the bibliography on how the metabolism and behavior of *K. marxianus* are affected after complete substrate consumption. This study aims to enrich the gaps, examining the growth and metabolism shifts of *K. marxianus* in a wide range of temperatures (20.0–35.0 °C) in increased lactose concentrations.

## 2. Materials and Methods

### 2.1. Microorganism and Growth Media

In the present study, *K. marxianus* strain EXF-5288 was isolated from kefir and kindly provided by Infrastructural Centre Mycosmo, MRIC UL, Slovenia. The yeast strain was conserved in YPDA medium (20 g/L glucose, 10 g/L yeast extract, 20 g/L peptone, and 20 g/L agar) at 4 ± 2 °C and for viability maintenance, it was sub-cultured before each experimental use. Cheese whey obtained from sheep milk was kindly offered by local cheese manufacturing (Hrysafis, Lemnos, Greece) and was frozen at −20 ± 2 °C until further use. Deproteinized cheese whey (DCW) was used as a carbon source, which was obtained by sterilization (121 °C, 20 min) followed by centrifugation (9000 rpm, 15 min, 4 °C) in a Universal 320R centrifuge (Hettich, Tuttlingen, Germany) and filtration. DCW contained approximately ~50.0 g/L lactose and a pH value of ~6.7. Further dilutions or concentrations were conducted to achieve initial lactose concentrations (35.0–100.0 g/L). Moisture (91.9%) and ash content (15.3%) was determined by AOAC standards [28]. Free amino acids nitrogen (FAN) was determined by Lie et al. [29] and was 29.0 mg/ L. Mineral salts were added into the media (concentration in g/L): KH_2_PO_4_ 7.00; Na_2_HPO_4_ 2.50; MgSO_4_·7H_2_O 1.50; FeCl_3_·6H_2_O 0.15; CaCl_2_·2H_2_O 0.15; ZnSO_4_·7H_2_O 0.02; and MnSO_4_·H_2_O 0.06. As a nitrogen source, urea was used at a concentration of 0.22% (*w*/*w*) [17].

### 2.2. Culture Conditions

Experiments in temperature values of (in °C) 20.0, 25.0, 30.0 and 35.0 were conducted under aseptic conditions (121 °C, 20 min) in a wide range of lactose concentrations (35.0–100.0 g/L). pH value of all cultures has been standardized prior to inoculation at 3.5 [17,18]. All cultures were performed in 250 mL Erlenmeyer flasks containing 50 ± 1 mL of sterilized growth medium. Media were inoculated with 1 mL (2% *v*/*v* inoculum) of exponential (24 h) pre-culture. The pre-culture was carried out in a YPD medium containing glucose ~20 g/L, peptone ~20 g/L and yeast extract ~10 g/L. All fermentations were carried out under fully aerobic conditions as flasks were placed in an orbital shaker (Labwit ZWY-211C, Melbourne, VIC, Australia) at an agitation rate of 180 ± 5 rpm. In fact, full aerobic conditions established during shake-flask experiments were proven through measurement of the concentration of dissolved oxygen (DOC, % *v*/*v*) carried out in the mentioned flask trials that had been performed. DOC was off-line measured using a selective electrode (HI 9146, Hanna Instruments, Woonsocket, RI, USA) according to the previously published procedure by Basa et al. [30] and Filippousi et al. [31].

To give more details concerning the performed determination, before harvesting the whole content of the flasks, in order to monitor the kinetics of the compounds, the shaker was stopped, and the probe measuring the concentration of oxygen was placed into the flask. Caution was paid in order for the liquid medium of the culture to cover all the measuring surfaces of the selective probe. Then, the shaker was switched on, and the measurement was taken after DOC equilibration (the mentioned equilibration was achieved within approximately 8–10 min after the probe had been inserted into the culture medium). The DOC was always ≥30% *v*/*v* (≥3.0 mg/L of dissolved oxygen) during all growth phases for all shake-flask experiments, providing evidence that fermentations were carried out under fully aerobic conditions [30,32,33]. Maintenance of pH value was achieved as described in Sarris et al. [34]. Scale-up experiments were conducted in a 6 L total volume bioreactor (Minifors 2, INFORS HT, Surrey, UK), with a 3 L working volume of sterilized growth medium at T = 20.0 °C and pH = 3.5. The bioreactor was equipped with a pH electrode (EasyFerm Plus PHI Arc 325, Hamilton, NY, USA) and oxygen probe (VisiFerm DO Arc 325, Hamilton, NY, USA). The media was saturated with oxygen (100% air-saturated) after sterilization. To keep the dissolved oxygen (DO) above 30% saturation, the agitation rate was set at 250 ± 5 rpm and airflow at 1 VVM.

### 2.3. Biomass and Single-Cell Protein Determination

Yeast cells were harvested by centrifugation (9000 rpm, 10 min, 4 °C) using a Universal 320R-Hettich centrifuge (Tuttlingen, Germany). The biomass was then washed with distilled water, and the supernatant was collected and stored at −20 °C for further analysis. Centrifugation was applied three more times under the same conditions. The biomass concentration was determined from dry weight (~85 °C until constant weight). The protein content of dry cells was determined by the Biuret method, as described in Dourou et al. [35]. Briefly, 10 mg of dry cells were initially disrupted by adding 1.0 mL of distilled water, 0.75 mL of KHPO_4_ 0.07 N and 3.0 mL of NaOH 5N. Heat treatment (100 °C) followed for 5 min. The concentration of protein content was determined with the assay of CuSO_4_.5H_2_O, and SCP was expressed as albumin equivalents at 540 nm (Shimadzu, Kyoto, Japan UV-1900 i).

### 2.4. Amino Acid Profile Determination

The method for amino acids (AAs) analysis was developed in KEAGRO AUTH, Greece. For AA analysis, 0.1 g of dry biomass was pretreated using 10 mL of 0.1 N HCl. Samples were centrifuged at 4000 rpm, 25 °C for 15 min. An amount of 100 μL of the supernatant was collected, and 900 μL MeOH was added. Aliquots were filtered with a 0.22 RC filter. The chromatographic determination of the amino acids (Appendix A) was carried out using an ExionLC/X500R (Sciex, Framingham, MA, USA) LC/QTOF system comprised of a vacuum degasser, two HPLC pumps, an autosampler, a column oven and a QTOF mass spectrometer. Separation was performed on a HILIC XBridge Amide, 100 × 2.1 (3.5 μm) column (Waters, Riga, Latvia). The mobile phase consisted of a binary mixture of 0.1% formic acid in water and 0.1% formic acid in ACN [0.1N HCl 90:10 (*v*/*v*)]. The flow rate was set at 0.50 mL/min, injection volume at 5 μL and column temperature at T = 40 °C. The sample was introduced into the mass spectrometer via electrospray ionization. The spray voltage was 5500 V, and the ion source gas 1 was set at 55 psi, and gas 2 was set at 50 psi. Curtain gas was set at 30 psi and ion source temperature at 550 °C. High-resolution-accurate mass data were obtained from 50–1000 Da in MS and MS/MS mode. The declustering potential was 50 V, and the collision energy was 35 V. The unit of the amino acid amount was converted to mg (amino acid)/g (dry biomass).

### 2.5. Evaluation of Ethanol and Organic Acids Production

Filtered aliquots of the collected supernatant culture medium, including ethanol, citric acid and acetic acid, were analyzed via HPLC (Appendix A). The HPLC system was equipped with a refractive index (RI) detector (RID-10A; Shimadzu Corp., Kyoto, Japan), online degasser (DGU-20A; Shimadzu Corp., Kyoto, Japan), high-pressure pump (LC-20AD; Shimadzu Corp., Kyoto, Japan), column oven (CTO-10ASvp; Shimadzu Corp., Kyoto, Japan), manual injector (7725i; Rheodyne, Rohnert Park, CA, USA, LLC), and a ReproGel H column (250 × 8 mm, 9 μm; Dr. Maisch, Ammerbuch, Germany). The column flow rate was 0.6 mL/min at T = 40 °C, while the mobile phase consisted of 5 mM H_2_SO_4_. The sample volume was 20 µL. Ethanol and organic acids were quantified using the standard calibration curves.

### 2.6. Statistical Analysis

Three independent cultures were conducted for all the trials performed, except the bioreactor trial, which was performed two independent times. Data were collected in Microsoft Excel and analyzed using XLSTAT software (Version, 2018.1., Addinsoft, Paris, France) via one-way analysis of variance (ANOVA). Mean comparisons were performed using Tukey’s HSD test adjustment at significance level α = 0.05 (*p* ≤ 0.05).

## 3. Results

### 3.1. Effect of Increased DCW Initial Lactose Concentrations in K. marxianus Strain EXF-5288 Growth

#### 3.1.1. Biomass and Single-Cell Protein Production

*K. marxianus* strain EXF-5288 growth was studied under increased initial lactose concentrations (50.0 ± 5.0, 75.0 ± 5.0 and 100.0 ± 10.0 g/L) at 35.0 °C and presented satisfactory growth in all trials and was able to assimilate the applied substrate. In trials with an initial DCW concentration of 50.0 ± 5.0 g/L, maximum biomass (X_max_) production did not exceed 4.24 ± 0.04 g/L (biomass yield on lactose consumed (Y_X/Laccons_) reached 0.08 ± 0.00 g/g) (Table 1). However, the growth of strain was favored by the presence of a higher initial DCW concentration (up to 75.0 ± 5.0 g/L). Specifically, maximum biomass production reached 9.40 ± 0.55 g/L with Y_X/Laccons_ = 0.12 ± 0.00 g/g in 168 h; however, further increase in initial substrate concentration up to 100.0 ± 10.0 g/L seems to not favor the biomass production which ranged between 3.63 ± 0.11–6.16 ± 0.07 g/L. Regarding SCP accumulation, maximum production (SCP_max_) reached 3.96 ± 0.74 g/L when the strain’s growth was monitored at 75.0 ± 5.0 g/L. In trials with DCW concentrations up to 50.0 ± 5.0 and 100.0 ± 10.0 g/L, SCP accumulation did not exceed 2.30 ± 0.07 and 2.74 ± 0.03 g/L, respectively. However, in all trials, the respective yield of protein of dry weight (Y_SCP/X_) ranged between 0.42–0.58 ± 0.03 g/g, indicating that different DCW initial concentrations do not affect the yield of protein content yet quantitative production.

#### 3.1.2. Ethanol and Organic Acids Production

*K. marxianus* strain EXF- 5288 presented the ability to produce and consume ethanol. An increase in initial DCW concentration led to the accretion of ethanol production, which was assimilated by the end of fermentations (Figure 1). In particular, maximum production of ethanol reached 35.5 ± 0.2 g/L with ethanol yield on lactose consumed (Y_Eth/Laccons_) of 0.35 ± 0.00 g/g in 72 h, in trials with initial DCW concentration up to 100.0 ± 10.0 g/L, while in trials with initial DCW concentration of 50.0 and 75.0 ± 5.0 g/L, ethanol production did not exceed 8.2 ± 0.0 and 18.2 ± 0.1 g/L, respectively (Table 1). In all of the flask experiments performed and for all growth steps of the mentioned cultures, DOC was off-line measured, and it was always found to be presented in values ≥30% *v*/*v*. At the first growth steps of all trials (0–30 h for the cases with initial lactose adjusted at 50.0 and 75 g/L, 0–72 h for the trial with initial lactose ≈100 g/L), the DOC values presented a significant and rapid drop, being, in any case, maintained in the values ≥30% *v*/*v*, suggesting the imposition of oxygen-sufficient (full aerobic) conditions. Thereafter, and during the step of ethanol oxidation, after lactose had been previously depleted (a step accompanied by further biomass production of the microorganism), the DOC values further increased to approximately 70% *v*/*v* for all trials performed. From the above-mentioned analysis, it can easily be suggested that *K. marxianus* EXF-5288 is a Crabtree-positive microorganism, capable of rapidly consuming sugar from the growth medium and converting it into ethanol, despite the aerobic conditions imposed into the medium, with the evident response of the mentioned biochemical behavior of the microorganism, the significantly higher production of ethanol (in g/L) compared to the production of dry yeast biomass (in g/L) when ethanol concentration peaked [32,36]. 

Moreover, as a typical event of most Crabtree-positive yeasts growing on sugars under oxygen-sufficient conditions, after sugar fermentation and (almost complete) assimilation from the medium, ethanol was re-consumed with a clearly lower consumption rate compared to the assimilation rate of sugar (see, i.e., Table 1). Ethanol was oxidized in order for either the cell energy of maintenance requirements to be generated, while in many cases and in accordance with the literature [32,36], further dry yeast biomass was synthesized through the so-called ethanol “make–accumulate–consume” strategy [32,36,37,38]. This phenomenon was clearly demonstrated in the present study (Table 1). Regarding organic acids production, it seems that citric acid production was not affected by the increase of initial DCW concentration since, in all trials, the production ranged between 2.0 ± 0.1–8.3 ± 0.1 g/L with a citric acid respective yield of the lactose consumed (Y_Cit/Laccons_): 0.03–0.12 ± 0.00 g/g. Moreover, traces of acetic acid were observed.

### 3.2. Effect of Different Culture Temperatures in K. marxianus Strain EXF-5288 Growth in DCW

#### 3.2.1. Biomass and Single-Cell Protein Production

In trials with an initial DCW concentration of 70.0 ± 5.0 g/L, strain represented satisfactory growth as well as maximum SCP production, as mentioned above. To study the strain’s ability to produce SCP under a wide range of temperatures, trials at 20.0, 25.0, 30.0 and 35.0 °C were conducted. Surprisingly, as shown in Table 2, lower temperatures favored biomass and SCP production. Specifically, maximum biomass production reached 14.24 ± 0.70 g/L with _YX/Laccons_ = 0.18 ± 0.00 g/g, under 20.0 °C, while statistical analysis did not show significant differences between trials of 20.0 and 25.0 °C. On the contrary, biomass production was not favored at higher temperatures since maximum values did not exceed 10.06 ± 0.40 (at 30.0 °C) and 8.55 ± 0.02 g/L (at 35.0 °C). Similar results were obtained regarding SCP production. Maximum production of SCP was observed during the strain’s growth at 20.0 °C, reaching SCP_max_ = 6.14 ± 0.66 g/L with respective SCP yield on a dry basis up to 0.43 ± 0.03 g/g. Similar to previous trials, the protein content of dry biomass was not affected by lower or higher temperatures.

#### 3.2.2. Ethanol and Organic Acids Production

Since the initial DCW concentration of substrate was 70.0 ± 5.0 g/L, and as was anticipated through the previous set of experiments (see Table 1), again, in the trials with the varied incubation temperatures imposed, production of ethanol was performed. As in the previous set of experiments, in the presently reported trials, cultures were performed under full aerobic conditions (DOC ≥ 30% *v*/*v*), while Eth_max_ values ranged between 13.9 ± 0.7–17.3 ± 0.9 g/L with Y_Eth/Laccons_ values being recorded = 0.20 ± 0.01–0.29 ± 0.03 g/g as shown in Table 2, while in all trials the strain was able to consume the produced ethanol after sugar had been depleted, resulting in further increase of yeast dry biomass production. Ethanol and citric acid production were not affected by temperature modifications. Specifically, in all trials, maximum values of citric acid did not exceed 6.2 ± 0.1–7.1 ± 0.4 g/L. Production of acetic acid was in traces in higher temperatures (30, 35 °C), reaching 4.8 ± 0.1 and 4.4 ± 0.2 g/L, respectively.

### 3.3. Effect of the Downgrade in DCW Initial Lactose Concentration in K. marxianus Strain EXF-5288 Growth

#### 3.3.1. Biomass and Single-Cell Protein Production

Regarding the above results and according to the previous analyses, it could be proposed that *K. marxianus* strain EXF-5288 is Crabtree-positive. Since respiration and ethanol fermentation are competitive pathways, downgrade lactose trials were conducted to avoid ethanol production (35.0 ± 2.0 g/L at 20.0 °C) (Figure 2). Maximum biomass production reached 9.69 ± 1.58 g/L with Y_X/Laccons_ = 0.28 ± 0.03 g/g in 120 h (Appendix A), compared to trials at 70.0 ± 5.0 g/L, where Y_X/Laccons_ reached 0.16 ± 0.00 g/g in the same hours of fermentation. On the contrary, lower lactose concentration in the applied substrate led to decreased SCP production since the maximum value did not exceed 3.66 ± 0.33 g/L with Y_SCP/X_ = 0.39 ± 0.03 g/g in 120 h.

#### 3.3.2. Ethanol and Organic Acid Production

Lower initial lactose concentration led to a significant decrease in ethanol production (Figure 2), whose maximum value did not exceed 2.8 ± 0.1 g/L in 24 h. Regarding the production of organic acids, the production of citric acid followed the same pattern as in previous trials, reaching Cit_max_ = 5.6 ± 0.0 g/L with Y_Cit/Laccons_ = 0.16 ± 0.00 g/g.

### 3.4. Effect of Upscale in Batch Bioreactor in K. marxianus Strain EXF-5288 Growth in DCW

Bioreactor trials were conducted at 65.0 ± 5.0 g/L under 20.0 °C since X_max_ and SCP_max_ values were observed under those conditions, as shown in Section 3.2.1. As shown in Table 3, the strain consumed the substrate within 48 h. Upscale conditions favored ethanol production compared to biomass production. Specifically, Eth_max_ value reached 19.2 ± 0.2 g/L with Y_Eth/Laccons_ = 0.34 ± 0.01 g/g, while in flask trials production of ethanol was 13.9 ± 0.7 g/L with Y_Eth/Laccons_ = 0.20 ± 0.01 g/g, under same conditions (Table 3). Moreover, biomass and SCP production were significantly decreased in bioreactor conditions compared to flask-shake trials at 70.0 ± 5.0 g/L lactose concentration since X_max_ and SCP_max_ did not exceed 3.63 ± 0.02 and 1.35 ± 0.05 g/L, respectively. However, the protein content of dry biomass reached 37.0% (*w*/*w*), indicating that the strain’s ability to accumulate single-cell protein is not affected by modifications of culture escalation. In bioreactor trials, citric acid production was slightly increased, reaching 10.5 ± 0.3 g/L (Y_Cit/Laccons_ = 0.18 ± 0.01 g/g) compared to the flask trials.

### 3.5. Amino Acid Profile

Amino acid profile analysis was conducted in all trials. Results showed that in all trials, the amino acid profile is the same, which proves that the composition of AAs is not affected by changes in temperature or by increased lactose concentrations. Specifically, six essential amino acids (EAA) were detected, including isoleucine, leucine, lysine, phenylalanine, valine, and threonine, while methionine, tryptophan, arginine and histidine were not detected. Non-essential amino acids (NEAA), aspartic acid, glutamine, glutamate (glutamic acid), glycine, proline, and serine were detected. The most abundant EAA in SCP was valine (9.5 mg/g), followed by leucine and isoleucine (4.0 and 3.9 mg/g, respectively). Glutamic acid’s amount was the highest of all the AAs, reaching 15.5 mg/g, followed by aspartic acid (12.0 mg/g).

## 4. Discussion

*Kluyveromyces marxianus* strain EXF-5288 was evaluated for its ability to grow on a wide range of DCW lactose concentrations under different temperatures (20.0–35.0 °C) in order to provide an insight into its metabolic behavior through the production of high-added-value metabolic compounds (Figure 3). *K. marxianus* strain EXF-5288 presented satisfactory growth in DCW substrate in all trials in shake-flasks. Specifically, in shake-flask cultures, maximum biomass and SCP production reached 14.24 ± 0.70 and 6.14 ± 0.66 g/L, respectively, under 20.0 °C. To the authors’ knowledge, this is the first report in the literature in which yeast *K. marxianus* is studied under 20.0 °C for SCP production. Since *K. marxianus* is thermophilic, most studies have focused on its growth under high temperatures. In the present study, different temperatures led to significant differences in biomass and SCP production; however, biomass yield was not affected by temperature alterations. This is in line with another study, where the formation of ethyl acetate by *Kluyveromyces marxianus* DSM 5422 depending on temperature (7.0 to 50.0 °C) was studied, resulting in no significant differences in biomass yield from 12.0 to 32.0 °C [39]. In the present study, upscale in a 3 L working volume bioreactor led to a decrease in biomass and SCP production, while the protein content of dry biomass ranged between 34.5–49.6% (*w*/*w*). Those results could be explained by the fact that the scale-up of fermentation processes is a multidisciplinary approach and can be affected by multiple factors, including agitation, mass transfer of oxygen, shear damage and others [40], resulting in the altered physiology of microorganisms with constant metabolic shifts, leading to reduced growth and productivity.

Yadav et al. [17] evaluated the SCP production by *K. marxianus* (in batch and continuous mode trials) and simultaneous COD removal using cheese whey under 40.0 °C (pH = 3.5, ~50.0 g/L lactose) in a 10 L bioreactor. Results showed that protein content reached 42.0% (*w*/*w*), which is in line with the findings of the present study regarding protein content. In another study, the potential of mixed culture of *K. marxianus* and *Candida krusei* to enhance COD removal efficiency and SCP production under extreme conditions (40.0 °C, pH = 3.5) was conducted in 10 L bioreactor [18]. The protein content of dry biomass was 43.4% *w*/*w*, comparable with the protein content range obtained from trials in the bioreactor in the present study.

*K. marxianus* strains are inherently characterized as Crabtree-negative without the ability to produce ethanol in the presence of glucose in the substrate [43]; however, *K. marxianus’s* overall metabolic pathway in glycolysis and the tricarboxylic acid (TCA) cycle and the dynamics of the involved gene transcripts remain unclear [22]. As a result, a plethora of studies have confirmed the ability of yeast *K. marxianus* to produce ethanol through bioprocesses [23], [44,45,46]. In this study, the increase of DCW lactose (50.0 ± 5.0, 75.0 ± 5.0, 100.0 ± 10.0 g/L) in the substrate increased ethanol production at constant temperature with values ranging between 8.2 ± 0.0–35.5 ± 0.2 g/L with Y_Eth/Laccons_ = 0.15 ± 0.00–0.35 ± 0.00 g/g, with all cultures being performed under full aerobic conditions, demonstrating a strong “Crabtree-positive behavior” for the employed yeast strain. The capability to ferment sugars into ethanol is a key metabolic trait of many (typical/“conventional”) yeast species/genera [36,47,48].

Crabtree-positive yeasts use fermentation even in the significant presence of oxygen in the growth medium [30,47], as indicated in the present investigation, where they could, in principle, rely on the respiration pathway. This is surprising because fermentation has a much lower ATP yield than respiration (2 molecules of ATP per glucose assimilated vs. approximately 18 molecules of ATP per molecule of assimilated glucose for Crabtree-positive yeasts) [47]. While the respiration pathway in eukaryotes may produce up to 37–38 molecules of ATP from one molecule of catabolized glucose (dependence on whether glucose enters inside the cell with the use of permease consuming one molecule of ATP per molecule of entering glucose, or the utilization of phosphotransferases), this value is substantially lower for Crabtree-positive microorganisms like *Saccharomyces cerevisiae*, presenting a low phosphate/oxygen (P/O) ratio; in *S. cerevisiae* and other Crabtee-positive yeasts, due to the low P/O value (in *S. cerevisiae* this value is ≈1.2), the yield for respiration accounts to *ca* 18 molecules of synthesized ATP per molecule of catabolized glucose [47], that, in any case, is significantly higher than the value recorded for fermentative assimilation of sugars [48]. 

In any case, the simultaneous monitoring of DOC values, the rapid lactose assimilation and the (appreciable in some cases) production of ethanol in the first (“fermentative”) step of the culture, despite the sufficient oxygen presence in the medium, easily suggests the categorization of the mentioned yeast strain as Crabtree-positive on lactose employed as the substrate. Further investigations related to the utilization of other sugars employed as carbon sources in order to identify their effect concerning the Crabtree effect, the threshold beyond which ethanol fermentation would cease, the cultivation of the microorganism under anaerobic conditions in order to identify and validate the potential Pasteur effect in this strain, and several assays of enzymatic activities in enzymes implicated in the central carbon catabolism of the microorganism would further validate the microbial behavior of the implicated strain, by in any case, the achieved kinetic results in both the flasks and the bioreactor evidence the implication of the Crabtree effect on the studied strain.

Concerning the production of ethanol by *K. marxianus* strain EXF-5288, the findings of the present study are partially in agreement with another study, where the effect of initial lactose concentration (42.0, 70.0, 100.0 and 150.0 g/l) on the fermentative performance of *K. marxianus* PTCC 5194 immobilized cells was studied in batch shake-flask experiments [49]. In the same study, the addition of lactose in cheese whey up to 70.0 g/L from 42.0 g/L enhanced ethanol production up to 23.6 g/L, while in the present study, lactose increase from 50.0 ± 5.0 to 75.0 ± 5.0 g/L led to an ethanol increase from 8.2 ± 0.0 to 18.2 ± 0.1 g/L. However, in the study of Roohina et al. [49], the increase of lactose concentration beyond 70.0 g/L significantly reduced ethanol production. Murari et al. [50] optimized bioethanol production from cheese whey using *K.marxianus* URM 7404 through response surface methodology, suggesting that the optimum conditions for ethanol production (>90% ethanol theoretical production from lactose conversion) were temperature ranging from 32.5 to 35.0 °C, pH 4.8 to 5.3, and lactose concentration 61.0 to 65.0 g/L, with temperature being the most significant factor. This is not in agreement with the findings of the present study since different temperatures did not lead to significant differences in ethanol production.

Biomass and SCP reduction in higher temperatures could be explained by the fact that although higher temperature stimulates respiration in the mitochondria of *K. marxianus,* it subdues the tricarboxylic acid (TCA) cycle, leading to a lowered ratio of reduced NADH)/NAD+ [51]. Regarding upscaling, strain *K.marxianus* EXF-5288 shifts its metabolism towards ethanol production with maximum value up to 19.2 ± 0.2 g/L with Y_Eth/Laccons_ = 0.34 ± 0.01 g/g, compared to shake-flask trials. Gabardo [52] studied the growth of immobilized cells of *K. marxianus* strains CBS 6556, CCT 4086, and CCT 2653 in batch fluidized-bed bioreactor indicating that initial lactose concentration of whey permeate (~60.0 g/L) was completely consumed, while yield of strain CCT 2653 consumed substrate was 0.33 g/g and strain’s CCT 4086 maximum ethanol production reached 28.0 g/L. Parrondo et al. [53] proposed a theoretical yield of 0.54 of ethanol per g of lactose for *K. marxianus* CECT 1123 since the ethanol yield of 0.51 g/g was obtained. However, in the present study, ethanol yield per lactose consumed did not exceed 35%, while SCP productivity ranged from 34 to 46% *w*/*w*. Furthermore, since ethanol yield was not affected under different temperatures and was significantly lower than theoretical yield, this study focused on SCP and biomass production.

Moreover, in the present study, there was an indication that *K. marxianus* strain EXF-5288 is able to consume ethanol when the applied substrate is consumed. Mo et al. [54] suggested two possible ethanol consumption routes in *K. marxianus,* with the first one is via cytoplasmic ADH6, which catalyzes ethanol to acetaldehyde, facilitated by NADP+, while the other one is by mitochondrial ATF1, which promotes ethanol esterification with the aid of acetyl-CoA. Sarris et al. [37] reported ethanol consumption after sugar exhaustion of the medium by *S. cerevisiae* MAK-1, indicating higher biomass production in the shake-flask experiments and to a lesser extent in the bioreactor trials under aeration, which is in line with this study. In the present study, the production of citric acid was achieved by *K. marxianus* EXF-5288, reaching 10.5 ± 0.3 g/L at bioreactor trials. Previous studies have reported the ability of *K. marxianus strains* to produce organic acids including lactic [23] and succinic acid [16]; however, the mentioned acids were not detected in the present study, and it could be hypothesized that their production is strain or substrate-dependent.

Previous studies have indicated that SCP is characterized by high nutritional value while it is rich in EAA [55,56,57]. However, the low content of sulfur-containing amino acids (i.e., methionine and cysteine) might be a limiting factor for the application of SCP in food systems [58]. Previous studies have reported low content or total absence of methionine in *K. marxianus* SCP [18,59,60], which is in agreement with the present study since methionine was not detected. On the contrary, the AAs profile in the present study was rich in glutamate (15.5 mg/g), aspartic acid (12.0 mg/g) and glutamine (9.5 mg/g), while regarding EAA valine was up to 9.5 mg/g. Similar results were observed in Yang et al. [61] study, where SCP derived from food waste (from one-stage fermentation) of yeast *Y. lipolytica* was rich in glutamate (10.6 mg/g), aspartic acid (7.5 mg/g) and valine (9.0 mg/g). In another report, when *K. marxianus* strain CBS 6556 utilized deproteinized cheese whey, SCP was rich in valine and leucine (7.5 and 7.7 g/100 g protein), while tryptophan was not detected [58]. Similarly, in the present study, EAA with higher concentrations were valine and leucine, while tryptophan was also not detected. Although many studies have investigated the SCP’s AAs profile, it has to be mentioned that amino acid production by fermentation processes is multifarious. Specifically, AAs production through bioprocesses is strongly influenced by many parameters such as inoculum size, pH, feed rate, temperature and oxygen transfer rate [62].

## 5. Conclusions

*K. marxianus* strain EXF-5288 presents satisfactory valorization of agricultural by-products such as cheese whey with simultaneous production of bio-compounds. Maximum SCP production at T = 20.0 °C by strain *K. marxianus* EXF-5288 in shake-flasks appears promising for a lower-cost energy process compared to its growth at higher temperatures (30.0 and 35.0 °C). However, reducing the energy costs of a biorefinery process is multifactorial and requires extensive investigation. In the present study, increased lactose concentrations led to increased ethanol production up to 35.5 ± 0.2 g/L, demonstrating the “Crabtree-positive” nature of strain EXF-5288, rendering *K. marxianus* a promising candidate for ethanol production. It could be suggested that utilization of lactose favored ethanol production in the preliminary stage of fermentations, and under the lack of carbon source strain, EXF-5288 consumed the produced ethanol. Therefore, the gene basis regarding ethanol consumption by *K. marxianus* strains demands further investigation. Furthermore, future work should emphasize the application of SCP derived from bioprocesses in the food sector (through food fortification, animal feed, etc.) and in the development of packaging materials while considering consumer perception and acceptance.

## Figures and Tables

**Figure 1 foods-13-01892-f001:**
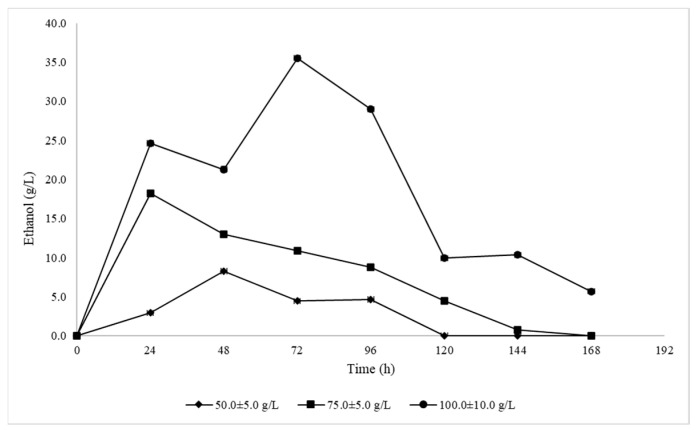
Ethanol production of *Kluyveromyces marxianus* strain EXF-5288 cultivated in increased DCW lactose concentration. Culture conditions: growth on aseptic 250 mL flasks at 180 ± 5 rpm, pH = 3.5, incubation temperature T = 35.0 °C.

**Figure 2 foods-13-01892-f002:**
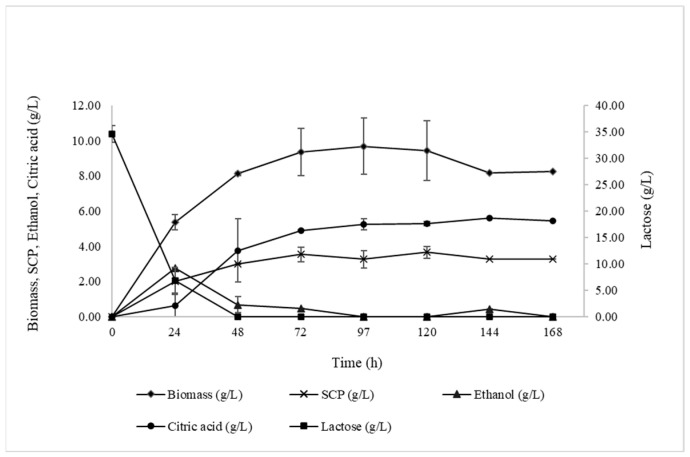
Biomass (g/L), SCP (g/L), Ethanol (g/L), Citric acid (g/L), and Lactose (g/L) evolution during growth of *Kluyveromyces marxianus* strain EXF-5288 on media with 35.0 ± 2.0 g/L initial DCW lactose concertation. Culture conditions: growth on aseptic 250 mL flasks at 180 ± 5 rpm, pH = 3.5, incubation temperature T = 20.0 °C.

**Figure 3 foods-13-01892-f003:**
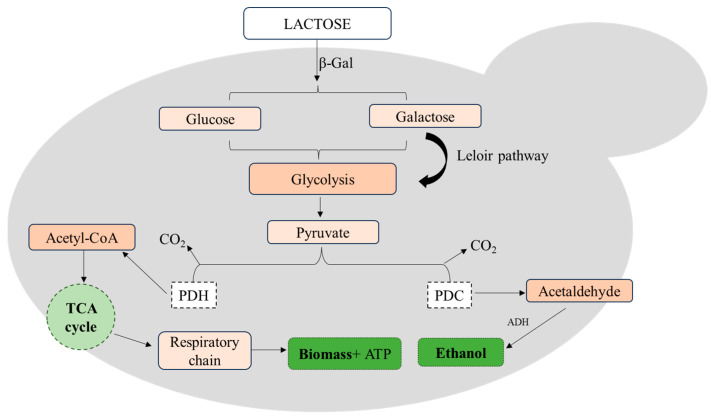
Pathways for utilization of lactose by *K. marxianus* with parallel production of metabolic compounds. Information about metabolic pathways is according to Martynova et al. [41] and Leonel et al. [42].

**Table 1 foods-13-01892-t001:** Kinetic data of *Kluyveromyces marxianus* strain EXF-5288 cultivated on increased DCW lactose concentrations under an incubation temperature of T = 35.0 °C.

Initial DCW Lactose (g/L)	Time (h)	Biomass, X (g/L)	SCP (g/L)	Laccons (g/L)	Ethanol (g/L)	Citric acid (g/L)	Y_Χ/Laccons_ (g/g)	Y_SCP/X_ (g/g)	Y_Eth/Laccons_ (g/g)	Y_Cit/Laccons_ (g/g)
50.0 ± 5.0	48 ^a,c^	4.24 ± 0.04 ^C^	1.96 ± 0.13	53.9 ± 0.1	8.2 ± 0.0 ^C^	6.1 ± 0.2	0.08 ± 0.00 ^B^	0.46 ± 0.03	0.15 ± 0.00 ^C^	0.11 ± 0.00
	120 ^b,d^	3.92 ± 0.01	2.30 ± 0.07 ^C^	53.4 ± 0.2	0.00	6.2 ± 0.1 ^C^	0.07 ± 0.00	0.58 ± 0.02 ^A^	0.00	0.12 ± 0.00 ^A^
75.0 ± 5.0	24 ^c^	6.00 ± 0.61	3.36 ± 0.81	73.3 ± 0.2	18.2 ± 0.1 ^B^	2.7 ± 0.0	0.08 ± 0.00	0.56 ± 0.02	0.25 ± 0.00 ^B^	0.04 ± 0.00
	120 ^d^	7.30 ± 0.10	3.30 ± 0.10	78.2 ± 0.1	4.4 ± 0.3	7.3 ± 0.2 ^B^	0.09 ± 0.00	0.45 ± 0.03	0.06 ± 0.00	0.09 ± 0.00 ^A^
	168 ^a,b^	9.40 ± 0.55 ^A^	3.96 ± 0.74 ^A^	78.2 ± 0.0	0.00	6.1 ± 0.1	0.12 ± 0.00 ^A^	0.42 ± 0.00 ^B^	0.00	0.08 ± 0.00
100.0 ± 10.0	72 ^c^	3.63 ± 0.11	1.95 ± 0.02	101.5 ± 0.8	35.5 ± 0.2 ^A^	4.2 ± 0.0	0.04 ± 0.00	0.54 ± 0.01	0.35 ± 0.00 ^A^	0.04 ± 0.00
	144 ^d^	4.03 ± 0.09	1.85 ± 0.02	110.0 ± 0.6	10.3 ± 0.2	8.3 ± 0.1 ^A^	0.04 ± 0.00	0.46 ± 0.01	0.09 ± 0.00	0.08 ± 0.00 ^A^
	168 ^a,b^	6.16 ± 0.07 ^B^	2.74 ± 0.03 ^B^	110.0 ± 0.2	5.7 ± 0.1	7.9 ± 0.1	0.06 ± 0.00 ^C^	0.45 ± 0.00 ^B^	0.05 ± 0.00	0.07 ± 0.00

^a^—max Biomass, ^b^—max SCP, ^c^—max Ethanol, ^d^—max Citric acid. SCP: Single-cell protein (g/L); Lac_cons_: consumed lactose (g/L); Y_X/Laccons_—Biomass yield on lactose consumed (g/g); Y_SCP/X_—Total SCP in dry basis (g/g); Y_Eth/Laccons_—Ethanol yield on lactose consumed (g/g); Y_Cit/Laccons_—yield on lactose consumed (g/g); DCW—Deproteinized cheese whey. Culture conditions: growth on aseptic 250 mL flasks at 180 ± 5 rpm, pH = 3.5. Means in a column representing maximum values, followed by the same letter (^A–C^), are not significantly different (*p* > 0.05).

**Table 2 foods-13-01892-t002:** Kinetic data of *Kluyveromyces marxianus* strain EXF-5288 cultivated on increased temperature, under initial lactose concentration 70.0 ± 5.0 g/L.

T °C	Time (h)	Biomass, X (g/L)	SCP (g/L)	Lac_cons_ (g/L)	Ethanol (g/L)	Citric Acid (g/L)	Y_Χ/Laccons_ (g/g)	Y_SCP/X_ (g/g)	Y_Eth/Laccons_ (g/g)	Y_Cit/Laccons_ (g/g)
20.0	48 ^c^	7.53 ± 0.62	3.08 ± 0.36	71.2 ± 0.8	13.9 ± 0.7 ^B^	2.3 ± 0.0	0.10 ± 0.00	0.41 ± 0.01	0.20 ± 0.01 ^A^	0.03 ± 0.00
	96 ^d^	11.13 ± 0.83	4.42 ± 0.37	75.9 ± 0.5	7.3 ± 0.8	6.8 ± 0.3 ^A^	0.14 ± 0.00	0.40 ± 0.01	0.10 ± 0.01	0.09 ± 0.00 ^A^
	144 ^a,b^	14.24 ± 0.70 ^A^	6.14 ± 0.66 ^A^	75.9 ± 0.5	0.00	5.0 ± 0.0	0.18 ± 0.00 ^A^	0.43 ± 0.03 ^A^	0.00	0.07 ± 0.00
25.0	24 ^c^	6.53 ± 0.10	2.88 ± 0.10	64.8 ± 7.7	17.1 ± 0.5 ^A^	2.1 ± 0.1	0.10 ± 0.01	0.44 ± 0.01	0.27 ± 0.04 ^A^	0.03 ± 0.00
	72 ^d^	8.79 ± 0.58	3.59 ± 0.21	69.9 ± 7.7	9.2 ± 1.0	6.9 ± 0.0 ^A^	0.13 ± 0.01	0.41 ± 0.01	0.13 ± 0.03	0.10 ± 0.01 ^A^
	144 ^b^	13.31 ± 0.66	5.45 ± 0.24 ^B^	69.7 ± 7.9	0.00	5.4 ± 0.5	0.20 ± 0.03	0.41 ± 0.04 ^A^	0.00	0.08 ± 0.02
	168 ^a^	13.80 ± 0.17 ^A^	5.31 ± 0.09	69.9 ± 7.7	0.00	6.1 ± 0.2	0.20 ± 0.02 ^A^	0.38 ± 0.01	0.00	0.09 ± 0.01
30.0	24 ^c^	5.74 ± 0.21	2.49 ± 0.11	68.2 ± 2.5	15.9 ± 0.4 ^A^	1.9 ± 0.0	0.08 ± 0.00	0.44 ± 0.03	0.23 ± 0.00 ^A^	0.03 ± 0.00
	120 ^d^	8.41 ± 1.14	3.47 ± 0.36	72.9 ± 2.5	4.0 ± 1.2	7.1 ± 0.4 ^A^	0.10 ± 0.00	0.41 ± 0.01	0.06 ± 0.02	0.10 ± 0.01 ^A^
	168 ^a,b^	10.06 ± 0.40 ^B^	4.11 ± 0.05 ^C^	72.9 ± 2.5	0.00	5.5 ± 0.0	0.14 ± 0.00 ^A^	0.41 ± 0.02 ^A^	0.00	0.08 ± 0.00
35.0	24 ^c^	5.39 ± 0.14	2.29 ± 0.11	60.8 ± 3.8	17.3 ± 0.9 ^A^	2.3 ± 0.1	0.09 ± 0.01	0.43 ± 0.04	0.29 ± 0.03 ^A^	0.04 ± 0.00
	120 ^d^	5.96 ± 1.54	2.64 ± 0.27	67.9 ± 1.7	3.2 ± 1.6	6.2 ± 0.1 ^A^	0.10 ± 0.01	0.46 ± 0.09	0.05 ± 0.02	0.09 ± 0.02 ^A^
	144 ^b^	7.88 ± 0.77	3.42 ± 0.20	67.95 ± 1.74	0.00	6.09 ± 0.02	0.12 ± 0.00	0.44 ± 0.09	0.00	0.09 ± 0.00
	168 ^a^	8.55 ± 0.02 ^B^	3.21 ± 0.17 ^C^	67.95 ± 1.74	0.00	6.05 ± 0.16	0.13 ± 0.00 ^A^	0.38 ± 0.09 ^A^	0.00	0.09 ± 0.00

^a^—max Biomass, ^b^—max SCP, ^c^—max Ethanol, ^d^—max Citric acid. Culture conditions: growth on aseptic 250 mL flasks at 180 ± 5 rpm, pH = 3.5. Means in a column representing maximum values, followed by the same letter (^A–C^), are not significantly different (*p* > 0.05).

**Table 3 foods-13-01892-t003:** Kinetic data of *Kluyveromyces marxianus* strain EXF-5288 cultivated at flasks and bioreactor, at T = 20.0 °C.

Culture Configuration	Time (h)	Biomass, X (g/L)	SCP (g/L)	Lac_cons_ (g/L)	Ethanol (g/L)	Citric Acid (g/L)	Y_Χ/Laccons_ (g/g)	Y_SCP/X_ (g/g)	Y_Eth/Laccons_ (g/g)	Y_Cit/Laccons_ (g/g)
70.0 ± 5.0 g/L Shake flasks	48 ^c^	7.53 ± 0.62	3.08 ± 0.36	71.2 ± 0.8	13.9 ± 0.7 ^B^	2.3 ± 0.0	0.10 ± 0.00	0.41 ± 0.01	0.20 ± 0.01 ^B^	0.03 ± 0.00
	96 ^d^	11.13 ± 0.83	4.42 ± 0.37	75.9 ± 0.5	7.3 ± 0.8	6.8 ± 0.3 ^B^	0.14 ± 0.00	0.40 ± 0.01	0.10 ± 0.01	0.09 ± 0.00 ^B^
	144 ^a,b^	14.24 ± 0.70 ^A^	6.14 ± 0.66 ^A^	75.9 ± 0.5	0.00	5.0 ± 0.0	0.18 ± 0.00 ^A^	0.43 ± 0.03 ^A^	0.00	0.07 ± 0.00
65.0 ± 5.0 g/L Stirred bioreactor	72 ^c,d^	3.37 ± 0.06	1.16 ± 0.02	57.1 ± 0.4	19.2 ± 0.2 ^A^	10.5 ± 0.3 ^A^	0.06 ± 0.00	0.34 ± 0.00	0.34 ± 0.01 ^A^	0.18 ± 0.01 ^A^
	120 ^a,b^	3.63 ± 0.02 ^B^	1.35 ± 0.05 ^B^	59.5 ± 0.3	17.4 ± 0.3	8.6 ± 0.4	0.06 ± 0.00 ^B^	0.37 ± 0.01 ^A^	0.29 ± 0.01	0.14 ± 0.01

^a^—max Biomass, ^b^—max SCP, ^c^—max Ethanol, ^d^—max Citric acid. Culture conditions: growth on aseptic 250 mL flasks at 180 ± 5 rpm and 3L bioreactor at 250 ± 5 rpm and 1 vvm, pH = 3.5. Means in a column representing maximum values, followed by the same letter (^A, B^) are not significantly different (*p* > 0.05).

## Data Availability

The original contributions presented in the study are included in the article/Appendix A, further inquiries can be directed to the corresponding author.

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
