# Peer review of "Single-Cell Protein and Ethanol Production of a Newly Isolated Kluyveromyces marxianus Strain through Cheese Whey Valorization"

_foods, 2024, doi:10.3390/foods13121892_

Round 1
Reviewer 1 Report
Comments and Suggestions for Authors
The study examines the ethanol production process of a recently discovered strain of Kluyveromyces marxianus and its metabolic behaviour in single-cell protein through cheese whey valorization.
Comments:
The authors must provide an explanation for their comparison between bioreactor trials cultivated at 65.0 g/L (DCW) and flasks cultivated at 35 and 70 g/L (DCW), instead of comparing with bioreactor trials at 35 and 70 g/L as well.
Discussion: Please give a more elaborate discussion on why the upscale in 3L working volume bioreactor led to decrease of biomass and SCP production; including how the protein content of dry biomass was affected.
Discussion: The authors must explain why other important organic acids (such as, succinic, lactic acid) were not quantified in this study
English needs proofreading. There are some grammar and typo mistakes.
Comments on the Quality of English LanguageEnglish needs proofreading. There are some grammar and typo mistakes.
Reviewer 2 Report
Comments and Suggestions for Authors
The present work evaluates ethanol and single-cell protein production from a strain of Kluyveromyces marxianus EXF-5288. Deproteinized cheese whey is used as a substrate and different temperature conditions are evaluated. The results are interesting, however before considering the work for publication the following aspects should be considered:
a) The rigorous experimental design used for the experiments is not presented, nor is a statistical analysis that validates the experimental results obtained.
b) An illustrative diagram of the metabolic pathway elucidated to produce the compounds formed is suggested.
c) Provide more information on the whey used. The composition of whey changes significantly depending on its origin.
d) The manuscript exhibits some grammatical errors and misspellings. It can certainly benefit from an extensive language revision.
Comments on the Quality of English LanguageNo comment
Reviewer 3 Report
Comments and Suggestions for Authors
In this manuscript, the authors comment on the use of a newly isolated Kluyveromyces strain; however, they do not describe the source of the isolation or any detail regarding the isolation. Additionally, the authors should be cautious about including "Metabolic behavior" in the title while the scientific evidence is not enough to describe it. In this regard, the title should be related to the manuscript description. Further, the authors affirm the “Crab-tree positive” nature of K. marxianus EXF-5288. However, a lot of experimental evidence is needed to support this affirmation, such as the oxygen concentration, oxygen uptake rate, oxygen mass transfer coefficient, glycolytic genetic expression, and enzyme activity, using glucose as a carbon source, etc.
The authors should provide a statistical analysis of the data to evidence the statistical differences of the experiments.
In Figure 1, the authors should remove the title legend "Ethanol (g/L)" in the graph area. In the Figures and Tables, the authors should provide the variable title "Time (h)" instead of "Hours".
In upscale batch bioreactor conditions, the authors should report the scale-up criteria and all oxygen parameters. Also, the authors should discuss the data from this point of view. In the bioreactor, there are different oxygen transfer kinetics compared with flasks. Additionally, the authors should provide details of the bioreactor set-up; was the media saturated with oxygen? Was the pH and temperature controlled? What were the dynamics of oxygen saturation, etc.? This data will contribute to the discussion of the results.
The authors should compare the productivity of SCP and theoretical ethanol yield in the experiments and discuss.
The authors should provide a proper discussion of the results with scientific fundamentals explaining the observations. In addition, the comparison with other studies should be concrete and related to the obtained results.
The authors should provide some arguments or the profile of temperature in the course of upscale batch bioreactor trials or energetic consumption/cost maintaining the fermentation at 20°C, supporting the conclusion that SCP production at T=20.0°C seems to be promising for a lower energy procedure in terms of cost.
In the conclusion, the authors should describe future work, scientific gaps, and possible applications of the generated knowledge.
Reviewer 4 Report
Comments and Suggestions for Authors
Authors have tried to optimize the production of single cell protein and ethanol form the Kluyveromyces marxianus strain. The manuscript need to be refined further for publication consideration.
1. The similarity index shows 22% overall similarity. Authors need to reduce the overall similarity under 10%.
2. The hplc chromatograms of the analyses carried out by the authors has to be included in the manuscript especially for Amino acid profile of SCP. Other chromotograms can be included as supplementary data.
3. Statistical analyses was not done to denote the significant difference between each trials. Thus it is not up to the standards of publication.
4. The side headings given by the authors for results section 3.1, 3.2 etc., are statements. Authors need to rewrite the article in a appropriate way to give concise side headings for better understanding.
Thus the manuscript may be considered for publication with above mentioned major revisions.
Comments on the Quality of English LanguageThe manuscript has to be edited on whole by a native English speaker. Especially the abstract of the manuscript is not up to the standards of publication.
Reviewer 5 Report
Comments and Suggestions for Authors
1. The molecular formula numbers in lines 93-94 on the third page and line 148 on the fourth page should be subscripted;
2. What is line 100 on the third page to adjust the pH to 3.5? What is the basis for this? Generally, the optimum pH for yeast growth is 4.5-5.0
3. On the third page, line 109, vvm is the abbreviation of ventilation ratio and should be capitalized.
4. Missing error bars in Figure 1
5. Is glycose in line 311 on page 10 misspelled? Should it be glucose?
6. This article mainly studies the effects of lactose concentration and temperature in DCW on the production of SCP and ethanol by K. marxianus strain EXF-5288. It is concluded that the optimal temperature is 20°C and the lactose concentration is about 70, but we know that yeast needs oxygen to grow. Yeast does not require oxygen to produce ethanol. Therefore, why did the author not study the effects of ventilation ratio or shaker speed or liquid volume on the production of SCP and ethanol by this strain? What is the reason for the large difference between the author's results in shake flask experiments and those in bioreactors? Is it the influence of oxygen? It is recommended that the authors add relevant discussions or supplement relevant experiments
Round 2
Reviewer 3 Report
Comments and Suggestions for Authors
In this revised version of the manuscript, the authors addressed the suggestions made in the first round of revision. However, there is still a suggestion for improving the manuscript.
Regarding the previous comment: "The authors should provide a proper discussion of the results with scientific fundamentals explaining the observations."
In lines 425-459, the authors describe many studies reporting different results. The comparison should be concise and should relate to the fundamentals explaining the phenomena. Responding to the scientific question of why temperature affects the production of ethanol, biomass, and SCP in Kluyveromyces, what changes in the metabolism of Kluyveromyces are related to the production of ethanol, biomass, and SCP at different temperatures, and the concentration of the carbon source is essential.
This should be similar to what the authors do in lines 460-468, explaining the ethanol consumption routes when the substrate is consumed, and in lines 391-424, relating the fundamentals of Crabtree behavior with the observations.
Reviewer 4 Report
Comments and Suggestions for Authors
The authors have addressed all the concerns raised by the reviewer. Thus the manuscript may be considered for publication.
Author Response
Comment:
The authors have addressed all the concerns raised by the reviewer. Thus the manuscript may be considered for publication.
Response:
Thank you. We greatly appreciate the time spent reviewing the manuscript and providing constructive and helpful comments.